# A Video-Game-Based Oral Health Intervention in Primary Schools—A Randomised Controlled Trial

**DOI:** 10.3390/dj10050090

**Published:** 2022-05-19

**Authors:** Ahmad Aljafari, Rawan ElKarmi, Osama Nasser, Ala’a Atef, Marie Therese Hosey

**Affiliations:** 1Department of Paediatric Dentistry, Orthodontics, and Preventive Dentistry, School of Dentistry, The University of Jordan, Amman 11942, Jordan; r.elkarmi@ju.edu.jo; 2Department of Dentistry, Jordan University Hospital, Amman 11942, Jordan; asa8200703@ju.edu.jo (O.N.); ala8201501@ju.edu.jo (A.A.); 3Centre of Oral, Clinical and Translational Science, Faculty of Dentistry, Oral & Craniofacial Sciences, King’s College London, London WC2R 2LS, UK; m.t.hosey@kcl.ac.uk

**Keywords:** preventive dentistry, oral hygiene, health education, dental caries, toothbrushing, healthy diet

## Abstract

Background: Poor oral health practices and high levels of dental caries have been reported among children in the developing world. Video games have been successful in promoting oral health in children. The aim of this study was to assess the impact of an oral-health-education video game on children’s dietary knowledge and dietary and toothbrushing practices; Methods: Two Schools in Amman, Jordan were randomly selected and assigned to either intervention or control. Six- to eight-year old children took part. The intervention group played the oral-health-education video game; the control group received no intervention. The groups were compared in terms of changes in: child dietary knowledge, dietary and toothbrushing practices, plaque scores, and parental familiarity with preventive treatments. Data were submitted to statistical analysis with the significance level set at *p* ≤ 0.05. Results: Two hundred and seventy-eight children took part. Most (92%) had carious teeth. At baseline, children reported having more than one sugary snack a day and only 33% were brushing twice a day. Most parents were unaware of fluoride varnish (66%) or fissure sealants (81%). At follow-up, children in the intervention group had significantly better dietary knowledge, and parents in both groups became more familiar with fluoride varnish. There were no significant changes in children’s plaque scores, toothbrushing and dietary practices, or parental familiarity with fissure sealants in either group. Conclusions: Using an oral-health-education video game improved children’s dietary knowledge. However, future efforts should target children together with parents, and need to be supplemented by wider oral-health-promotion.

## 1. Introduction

Dental caries affects a significant number of young children across the globe [1]. Children in developing countries and those living with poor socio-economic circumstances in developed countries are more likely to be affected [2]. Jordan as a developing country, is no exception. A previous study revealed that 76% of six-year-old and 46% of twelve-year-old children in Jordan have dental caries [3] with no significant reduction over the last few decades [4].

Caries is a multifactorial disease that can be influenced by oral hygiene and dietary practices. Cariogenic foods and drinks are regularly consumed by most children in Jordan [5]. Only half report brushing their teeth once or more a day. In addition, dental attendance is poor, and usually only when in pain [6]. Findings of a recent study suggest that lack of awareness of evidence-based dietary and oral hygiene practices is prevalent in Jordanian families [7]. In fact, the majority of parents do not recognise the importance of fluoride in toothpaste or the reduction of sugar intake frequency, and many had difficulties in identifying sources of sugar in their diet [7].

It is clear that children in Jordan are in need of oral health promotion. The World Health Organization’s (WHO) Ottawa Charter recommends the delivery of health education as part of health promotion [8], and knowledge remains an essential component needed to achieve behaviour change [9]. Video games are easy to deliver, require less manpower, and have been successfully used to deliver education on a variety of education and health-related topics [10,11]. An oral health education video game evaluated at a hospital was found to be as effective as a dental health educator in improving children’s recognition of healthy foods and drinks, and in reducing children’s sweetened drink consumption [12,13]. However, waiting until children attend for dental care to deliver oral health education is often too late, especially in countries and populations that attend only when in pain. As such, there is a need to explore whether video games can contribute to oral health education in other, non-clinical, settings.

Most schools in Jordan have computer rooms with internet access, and 62% of the pupils reported that they have used those facilities during their education [14]. In addition, 75% of Jordanians own a smartphone, and 80% have access to the internet at their homes [15]. The aim of the current study was to evaluate whether introducing an oral health education video game at schools can improve children’s dietary knowledge and dietary and toothbrushing practices, and parents familiarity with preventive treatments including toothpaste, fluoride varnish, and fissure sealants. As such, the null hypothesis for this study was that playing an oral health education video game does not lead to significant changes in children’s dietary knowledge, dietary habits, oral hygiene habits, or parental familiarity with preventive treatments.

## 2. Materials and Methods

### 2.1. Trial Design

This was a two-armed prospective Randomised Controlled Trial (RCT). Two schools in Amman, Jordan were randomly selected then assigned into two groups:Intervention group: children played an oral health education video game in school, and were given instructions to play the game at home.Control group: Received no intervention.

The study was registered in the ISRCTN registry (registration number: ISRCTN16292972, registration date: 25/02/2020). The Consolidated Standards of Reporting Trials (CONSORT) were applied to ensure thorough design and reporting. Procedures followed were in accordance with the Helsinki Declaration of 1975, as revised in 2000. Ethical approval was granted by Jordan University Hospital’s Institutional Review Board (reference number: 2019/176). Written parental consent was sought and both children and parents were informed that they did not have to answer any questions and could withdraw from the study at any time without giving an explanation and without any impact on their care.

### 2.2. Participants

The target population was children in the first and second grade (six- to eight-years-old) in Amman, Jordan. The inclusion criteria for the schools taking part were to: (i) be a school in Amman, Jordan, (ii) have both boys and girls, and (iii) have at least 120 students in first and second grade. The inclusion criteria for children were: (i) to be in first or second grade and (ii) to not suffer from a diagnosed learning disability. Children absent from school on the day of data collection or whose parents declined to take part in the study were excluded.

### 2.3. Interventions

Children in the intervention group played an oral health education video game in their school’s computer lab under the supervision of researchers ON and AAT. They were also given written and verbal instructions to download the game at home. Children in the control group did not receive any intervention. 

The video game was based on a previously developed English-speaking game [12]. The oral health education content was based on Public Health England’s (PHE) ‘delivering better oral health’ [16] and included recommending twice daily toothbrushing with at least a 1000ppm fluoride toothpaste, reducing sugar frequency, promoting the ‘Eatwell Plate’ as well as promoting regular dental visits and the application of fluoride varnish and fissure sealant to the permanent molars of high caries risk children. The game maintained most of the ‘script’ of the English-speaking game. However, some changes and upgrades were necessary; translation into Modern Standard Arabic (MSA), developing new animations and graphics, changing some food and drink items to make them culturally suitable, further development of the original game’s ‘toothbrushing’ and ‘visiting the dentist’ sections, and finally, the addition of in-game rewards to positively reinforce healthy choices. 

The developed prototype was then piloted along with the study’s outcome measures in a convenience sample of seventeen children and parents attending the dental department at a large public hospital. Participants would play the game on a smart device and examine the study’s outcome measures, and then would provide feedback. The researchers recorded this feedback in a notebook then used simple content analysis to note the participants’ opinions and refine the video game and the study’s outcome measures. Figure 1 displays some parts of the game.

### 2.4. Outcomes

#### 2.4.1. Primary Outcome Measures

Changes in children’s dietary knowledge recorded using a Pictorial Dietary Questionnaire (PDQ) [12]. The PDQ contains 70 food and drink items that the children are asked to classify into ‘healthy’ and ‘unhealthy’.

#### 2.4.2. Secondary Outcome Measures

Changes in dietary and toothbrushing practices reported by children using toothbrushing and diet diaries.Changes in dietary practices reported by parents using the Child Dietary Questionnaire (CDQ) [17]. The CDQ measures dietary intake in four domains: Fruits and vegetables, sweetened drinks, non-core foods, and fat from dairy.Changes in children’s toothbrushing habits, and changes to parental recognition of fluoride toothpaste concentration and in-office preventive treatments, recorded using a parental questionnaire.Changes in children’s plaque scores using the Simplified Oral Hygiene Index (S-OHI) [18].Number of children downloading and playing the game at home recorded using Google and Unity developer tools.

#### 2.4.3. Data Collection

At baseline, children in both groups completed the PDQ under the researchers’ supervision to assess their dietary knowledge. They were also given diet and toothbrushing diaries, and a parental questionnaire to complete at home and return to the researchers. In addition, researcher (AA) performed an oral examination for children in both groups and recorded their dmft and plaque score. Three months after the intervention, the schools were revisited. Children from both groups were re-examined to note their plaque scores and the children completed the PDQ, diet and toothbrushing diaries, and parental questionnaire once again.

### 2.5. The Dental Examination

The examination took place in a well-lit room and the recommendations outlined by the WHO’s manual were followed [19]. The examiner (AA) noted the child’s decayed, missing due to caries, and filled teeth in primary dentition (dmft). Caries was recorded at the ‘obvious decay experience’ level, defined as ‘caries that can be visualised through the enamel or lesions where it has advanced to form a frank cavity’. No radiographs were taken for caries detection. The examiner also recorded the children’s plaque score according to the S-OHI [17]. To ensure examiner calibration, AA underwent a pilot process by examining and comparing dmft and plaque score results in a sample of six children with another well-experienced examiner (RE) prior to the commencement of the trial. Inter-rater Kappa scores were 0.88 for caries and 0.70 for plaque score. Intra-rater Kappa score for caries for examiner AA was 0.94.

### 2.6. Sample Size

Based on the results of a previous study [13], the primary outcome measure’s (PDQ) expected mean score is 54.7, the expected standard deviation is 8, and the expected mean difference is 5.0. As such, a study with 95% power will require a total sample of 136 participants (68 per group) with completed datasets to compare between groups using a two-tailed independent samples *t*-test at 5% level of significance.

### 2.7. Randomisation and Blinding

A list of all schools in Amman fitting the inclusion criteria was obtained (121 schools). Computer-generated numbers were used to randomly select two schools to be invited to take part. The two schools were randomised into control and intervention groups using a simple computer-generated randomisation number. Researcher AA, who was responsible for performing the dental examination and contributed to data analysis, was blinded to group allocation. Only after data analysis was completed were group allocations revealed.

### 2.8. Statistical Methods

All data were entered into SPSS 20. Descriptive statistics for all explanatory variables were recorded and provided overall and by study group. Shapiro-Wilk test was used to determine data normality for each continuous variable. An Independent Samples Student *t*-test was used to compare the two groups whenever a normal distribution was present, while a Mann-Whitney-U test was used when the data did not follow a normal distribution. Paired Samples Student *t*-tests were used to compare baseline and follow-up scores for each group whenever a normal distribution was present. Chi-square test was used to assess differences in categorical variables between groups. Related Samples McNemar test was used to assess changes in categorical variables between baseline and follow-up. The significance level for all tests was set at *p* ≤ 0.05. Data analysis was performed according to the intention-to-treat principle; data for all those assigned to either group were included. Outcome variables with more than 5% of values missing at follow-up were handled using a five-step multiple imputation.

## 3. Results

### 3.1. Recruitment

Data collection took place between November 2019 and March 2020. On 15 March 2020, the Jordanian government shut down schools for the remainder of the school year due to the COVID-19 pandemic. A total of 278 children were recruited (143 in the intervention group, 135 in the control group). Figure 2 details recruitment and data collection by group.

### 3.2. Sample Description

The participants’ basic characteristics can be seen in Table 1. The children’s mean age was 6.5 years (SD = 0.5). Dental examinations were performed for 244 children (88%) at baseline. Two hundred and fourteen (88%) had caries in at least one primary tooth. The mean dmft score was 5.1 (range: 0–12, SD = 3.0). Parents of 223 children (77%) completed parental questionnaires at baseline. The mean age of mothers was 35.1 (range: 21–50, SD = 6.35) while the mean age of fathers was 41.8 (range: 28–75, SD = 7.65) years.

### 3.3. Children’s Dietary Knowledge

All children assigned to either group completed the PDQ at baseline. The mean score of children in the intervention group was 56.5 out of 70 (Range: 38–66, SD = 5.9), while for the control group it was 57.2 (Range: 41–66, SD = 4.9). Three months after the intervention, 261 children (130 Intervention, 131 Control) re-took the PDQ. Mean post-intervention PDQ score was 57.8 (Range: 36–68, SD = 6.3) in the intervention group, and 57.5 (Range: 41–66, SD = 5.1) in the control group. The improvement in the dietary knowledge of children in the intervention group was statistically significant (Paired samples *t*-test *p* = 0.019 *; 95% CI = 0.21–2.35), unlike the control group (Paired samples *t*-test *p* = 0.60). An independent samples *t*-test to compare the groups post intervention showed no statistically significant differences (*p* = 0.12).

### 3.4. Child-Reported Dietary Practices

Two hundred and five children (74%) completed three-day dietary diaries at baseline. Children in the intervention group had on average 1.2 sugary snacks a day (Range: 0–6, SD = 0.9), while those in the control group had 1.6 sugary snacks a day (Range: 0–6, SD = 1.1). A total of 147 children (69 intervention, 78 control) completed dietary diaries at the three-month follow-up. Children in the intervention groups reported having 1.1 sugary snacks a day post-intervention (Range: 0–3, SD = 0.9), while those in the control group reported having 1.3 (Range: 0–6, SD = 1.0). There were no statistically significant difference between the groups’ scores at follow-up (Independent samples Mann-Whitney-U test *p* = 0.12).

### 3.5. Parent-Reported Dietary Intake

Two hundred and twenty-three parents (72%) completed the CDQ at baseline, detailing the child’s food intake. One hundred and seventy-one parents (80 intervention, 91 control) re-completed the CDQ at the three-month follow-up. Independent samples *t*-tests revealed that there were no statistically significant differences between the groups in terms of food intake at follow-up.

Table 2 summarises the participants’ dietary knowledge and practices before and after the intervention and notes the recommended CDQ score in each food category [17].

### 3.6. Children’s Plaque Score

Two hundred and forty four children (88%) had their plaque score recorded at baseline and at the three-month follow-up. Mean plaque score at baseline was 1.16 (Range: 0.50–2.30, SD = 0.30) in the intervention group and 1.09 (Range: 0.20–2.20, SD = 0.39) in the control group. At the three-month follow-up, mean plaque score was 1.17 (Range: 0.50–2.00, SD = 0.29) for the intervention group and 1.21 (Range: 0.33–2.00, SD = 0.35) for the control group. A Mann-Whitney U test was used to compare the groups in terms of plaque score change at follow-up and revealed no statistically significant differences (*p* = 0.08).

### 3.7. Child-Reported Toothbrushing Practices

One hundred and ninety-one children (69%) completed toothbrushing diaries at baseline. The children’s mean toothbrushing frequency was 1.7 times a day (range: 0–3, SD = 0.44) in the intervention group, and 1.3 times a day (range: 0–3, SD = 0.74) in the control group. One hundred and forty children (54 intervention, 86 control) completed toothbrushing diaries at the three-month follow-up. Mean toothbrushing frequency reported was 1.6 times a day (range: 0–3, SD = 0.47) in the intervention group, and in the control group 1.4 times a day (range: 0–3, SD = 0.80). A Mann-Whitney U test confirmed no statistically significant differences between the two groups in toothbrushing frequency at follow-up (*p* = 0.81).

### 3.8. Parent-Reported Toothbrushing Practices

Two hundred and eighteen parents (78%) completed a questionnaire on the child’s oral hygiene habits at baseline. One hundred and sixty-five parents (78 intervention, 87 control) re-took the questionnaire at the three-month follow-up. At follow-up, there were no statistically significant differences between the two study groups in terms of toothbrushing frequency (Chi-square *p* = 0.65), parental supervision (Chi-square *p* = 0.51), or knowledge of fluoridated toothpaste concentration (Chi-square *p* = 0.14).

### 3.9. Parent’s Knowledge of Preventive Therapies

Two hundred and nineteen parents (79%) answered questions regarding their familiarity with caries preventive treatment provided at the dental office at baseline. One hundred and sixty-seven parents (78 intervention, 89 control) re-answered the questionnaire at the three-month follow-up. There was a statistically significant increase in the percentage of parents reporting familiarity with fluoride varnish at follow-up in both groups (intervention group Related Samples McNemar test *p* = 0.035 *, Control group Related Samples McNemar test *p* = 0.019 *). There was also a significant improvement in the control group’s familiarity with fissure sealants (Related Samples McNemar test *p* = 0.002 *).

Table 3 summarises changes in children’s oral hygiene practices and parents’ familiarity with preventive treatments before and after the intervention.

### 3.10. Video Game Usage Patterns

One hundred and ten parents (77%) in the intervention group answered a question regarding the availability of a smart device at home for the child to play the video game intervention. Ninety-four of them (86%) indicated that a device is available. However, only 39 participants (27%) downloaded the game at home as noted by Google Play’s developer console. Using Unity Developer tools it was noted that children spent a mean of 8.20 min (range = 0–18.32, SD = 5.15) playing the game every day in the first month, 4.44 min (range = 0–20.18, SD = 6.30) daily in the second month, and then 4.40 min (range = 0–15.28, SD = 5.47) daily in the third month.

## 4. Discussion

Playing an oral health promotion video game at school improved the children’s dietary knowledge, as evident by their improved pictorial dietary questionnaire scores on follow-up. Participating in the study improved parental familiarity with fluoride varnish and fissure sealants. The study sample is representative of children in public schools in Jordan and the findings highlight the high prevalence of caries and poor oral health practices in this population. Children participating in the study were eating more than one sugary snack a day and consuming more sweetened drinks and non-core foods and fewer fruits and vegetables than recommended [17]. Only a third were brushing their teeth twice a day and most were unsupervised. Their parents were not sure of the correct fluoride toothpaste concentration or the benefits of fissure sealant and fluoride varnish. Although the introduction of a standalone video game led to some improvement in children’s dietary knowledge, it did not lead to significant improvements in toothbrushing, dietary habits, or plaque scores, suggesting that this type of intervention might be best provided within wider oral health promotion programmes that include other elements such as school toothbrushing [20] and fluoride varnish application [21]. Furthermore, the importance of socioeconomic factors must be acknowledged. Our sample constituted of public school children in an urban area of a developing country. ‘Upstream’ action tackling socioeconomic determinants remains necessary [22].

Our findings confirm the results of a previous study that used an earlier version of the game [13], and are in line with the results of other studies investigating video game-based nutritional interventions [23,24] and smart phone application-based oral health interventions [25,26]. Improvements in children’s dietary knowledge in this study were lower than those reported previously [13]. This is most likely because follow-up in the current study was significantly longer, although the difference in population and the change in setting from hospital to school could also be contributing factors. An improvement in parental familiarity with fluoride varnish was noted at follow up. This was likely due to the dentist visit itself motivating parents to learn more in terms of oral health. It is also possible that parents learned from re-doing the questionnaire, or that they were simply giving a socially desired answer [27].

In a previous study, dentists cited parental motivation as a key barrier to preventive care delivery [28]. This underlines the importance of involving parents ‘directly’ in future oral health promotion efforts. In our study, we could not measure how often the parents played the game with their child. A previous study has found that families with less parental educational attainment had reduced engagement with a standalone home care computer game [29]. The COVID-19 pandemic caused an acceleration towards online learning in Jordan and the world. This highlights the importance of developing digital, evidence-based, child-friendly oral health education that can be incorporated into schools’ online curriculums; however, our findings also raise questions whether some less affluent families can engage with such learning opportunities or are at risk of being left behind and missing schoolteacher support.

Children need capabilities, including knowledge, opportunity, and motivation in order to achieve behaviour change [9]. Successful oral health promotion programmes incorporate a variety of interventions [30]. As such, providing a healthy school environment is needed for the success of future programmes in Jordan and similar developing countries. Video games can be used to provide knowledge and motivation on oral disease prevention in children and individuals under high risk, such as those undergoing orthodontic treatment, but they need to be part of wider approaches that advocate for stricter control of cariogenic foods in public schools and better in-school health facilities that can support toothbrushing programmes and fluoride varnish application.

This study had its limitations. First, we had planned to roll out the study to a number of different schools, but data collection was interrupted by COVID-19-related school closures. Second, some findings of this study could be due to the participants merely taking part in the study or coming into contact with the research team [27]. Third, some measures were self-reported, meaning that some might have given socially desired answers. Nonetheless, we used a dietary questionnaire that was recommended for the targeted age group [31] and there is some evidence that child-reported diaries are reliable [32]. Forth, study groups were randomly assigned on school level and not on participant level. However, this was important since assigning groups on a participant level would have risked sample contamination within schools. Finally, it would have been beneficial to track video game usage patterns by participant. However, that would have required participants to have usernames and passwords to log in; a step that we felt might deter some parents and children from using the game at home.

The study also had its strengths. First, care was taken during sample selection and blinding. Second, recruited families largely represent the population in terms of education and employment [33]. Finally, as far as we are aware, this is the first evaluation of an evidence-based Arabic-language-speaking oral health video game.

## 5. Conclusions

Using a standalone oral health education video game in school leads to improved dietary knowledge in children that is retained for at least three months but does not result in behaviour changes in terms of dietary or oral hygiene practices. Future efforts should target children in settings where they are accompanied by parents and the oral health education delivered needs to be supported by wider oral health promotion.

## Figures and Tables

**Figure 1 dentistry-10-00090-f001:**
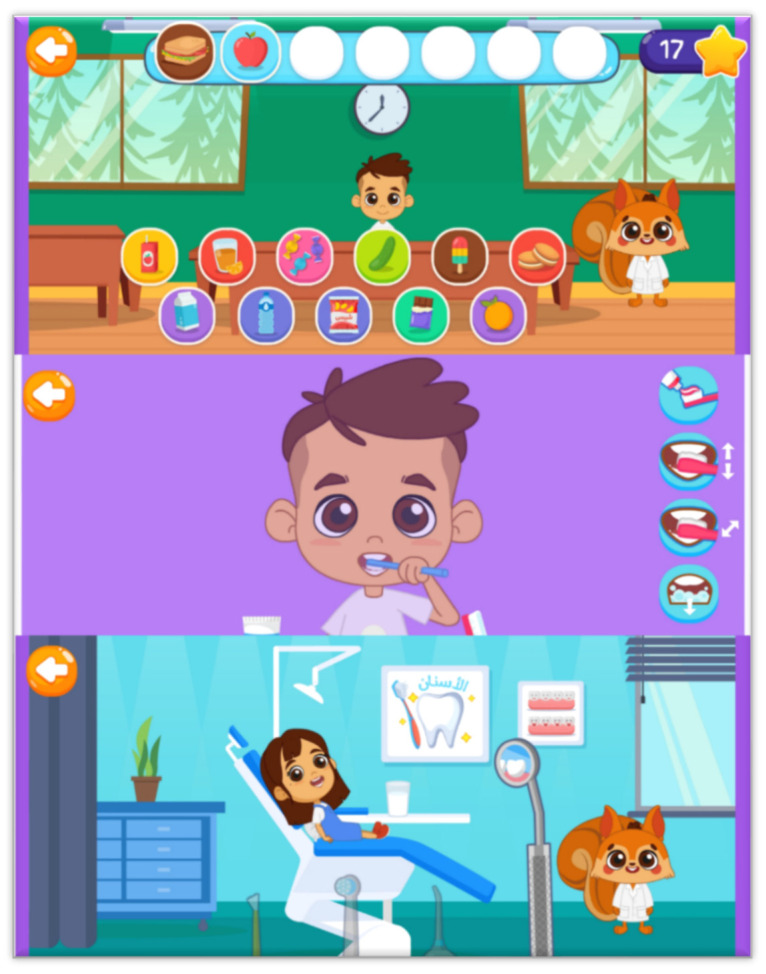
The video game.

**Figure 2 dentistry-10-00090-f002:**
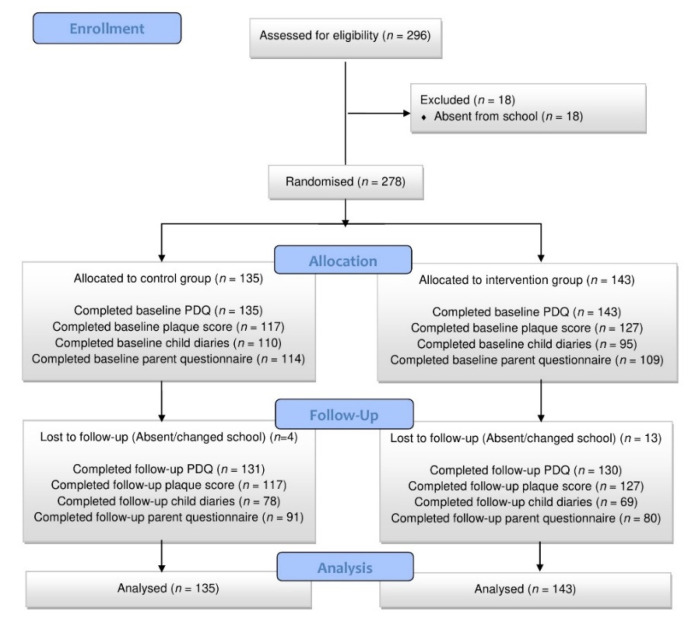
Recruitment flowchart.

**Table 1 dentistry-10-00090-t001:** The participants’ basic characteristics.

	Intervention Group	Control Group
	Mean ± SD	Mean ± SD
* **Age** *	6.5 ± 0.5	6.5 ± 0.5
* **dmft** *	5.3 ± 2.9	4.8 ± 3.0
* **Mother’s age** *	34.8 ± 6.6	35.4 ± 6.1
* **Father’s age** *	41.8 ± 8.1	41.8 ± 7.3
	*n* (%)	*n* (%)
* **First grade** *	69 (48%)	65 (48%)
* **Second grade** *	74 (52%)	70 (52%)
* **Male** *	77 (54%)	65 (48%)
* **Female** *	66 (46%)	70 (52%)
* **Both parents unemployed** *	12 (11%)	10 (9%)
* **At least one parent employed** *	95 (89%)	98 (91%)
* **Mother attended primary school only** *	32 (30%)	12 (11%)
* **Mother attended secondary school or higher** *	76 (70%)	96 (89%)
* **Father attended primary school only** *	40 (38%)	14 (13%)
* **Father attended secondary school or higher** *	66 (62%)	94 (87%)

**Table 2 dentistry-10-00090-t002:** Participants’ dietary knowledge and practices before and after the intervention.

Domain			Baseline Mean ± SD	Follow-Up Mean ± SD	*p*-Value
* **Dietary knowledge** *	**PDQ score**	Control	57.2 ± 4.9	57.5 ± 5.1	0.60
		Intervention	56.5 ± 5.9	57.8 ± 6.3	0.019 *
		Intergroup difference at follow up *p* = 0.12			
* **Dietary practices** *	**Number of sugary snacks**	Control	1.6 ± 1.1	1.3 ± 1.0	0.20
		Intervention	1.2 ± 0.9	1.1 ± 0.9	0.92
		Intergroup difference at follow up *p* = 0.12			
	**Fruits and vegetables** **(recommended ≥ 14)**	Control	13.4 ± 5.5	14.7 ± 5.3	0.06
		Intervention	13.6 ± 5.9	14.1 ± 6.9	0.61
		Intergroup difference at follow up *p* = 0.36			
	**Fat from dairy** **(recommended = 0)**	Control	6.7 ± 5.2	7.5 ± 4.9	0.25
		Intervention	7.6 ± 4.9	7.7 ± 5.3	0.95
		Intergroup difference at follow up *p* = 0.46			
	**Sweetened drinks** **(recommended ≤ 1)**	Control	2.5 ± 1.7	2.4 ± 1.7	0.84
		Intervention	2.6 ± 1.8	2.5 ± 1.7	0.62
		Intergroup difference at follow up *p* = 0.81			
	**Non-core foods** **(recommended ≤ 2)**	Control	2.7 ± 1.3	2.8 ± 1.7	0.65
		Intervention	3.1 ± 1.5	3.3 ± 1.6	0.57
		Intergroup difference at follow up *p* = 0.91			

* *p* ≤ 0.05.

**Table 3 dentistry-10-00090-t003:** Oral hygiene practices and parental familiarity with preventive treatments.

**Domain**			**Baseline**	**Follow-Up**	
			**Mean ± SD**	**Mean ± SD**	***p*-Value**
**Oral hygiene practices**	Plaque score	Control	1.09 ± 0.39	1.21 ± 0.35	0.01 *
		Intervention	1.16 ± 0.30	1.17 ± 0.29	0.29
		Intergroup difference at follow up *p* = 0.08			
	Brushing frequency	Control	1.3 ± 0.74	1.4 ± 0.72	0.60
		Intervention	1.7 ± 0.44	1.6 ± 0.47	0.65
		Intergroup difference at follow up *p* = 0.81			
			*n* (%)	*n* (%)	
	Child brushes twice or more daily	Control	32 (29%)	35 (41%)	0.14
		Intervention	40 (37%)	32 (41%)	0.99
		Intergroup difference at follow up *p* = 0.65			
	Child brushes supervised	Control	17 (16%)	8 (10%)	0.13
		Intervention	15 (14%)	12 (15%)	0.99
		Intergroup difference at follow up *p* = 0.51			
	Parent not sure of correct toothpaste concentration	Control	71 (65%)	56 (64%)	0.82
		Intervention	64 (59%)	42 (55%)	0.06
		Intergroup difference at follow up *p* = 0.14			
* **Parental familiarity with preventive treatments** *	Parent familiar with fluoride varnish	Control	38 (35%)	44 (49%)	**0.019** *
		Intervention	36 (33%)	39 (50%)	0.035 *
		Intergroup difference at follow up *p* = 0.94			
	Parent familiar with fissure sealants	Control	19 (17%)	30 (65%)	0.002 *
		Intervention	23 (21%)	17 (22%)	0.99
		Intergroup difference at follow up *p* = 0.001 *			

* *p* ≤ 0.05.

## Data Availability

The datasets used and/or analysed during the current study are available from the corresponding author on reasonable request.

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
