# Peer review of "A Video-Game-Based Oral Health Intervention in Primary Schools—A Randomised Controlled Trial"

_dentistry, 2022, doi:10.3390/dj10050090_

Round 1
Reviewer 1 Report
Thank you for submitting this manuscript for review. This is an exciting topic and gamification of education is a newer tool that we can use in oral health education but we need to test how it can be beneficial.
Methods:
Section 2.8, it might be beneficial to state that no radiographs were made for caries detection. You do state that it was visual inspection.
Discussion:
Line 283-285, “It is mentioned that playing an oral health promotion video game at school improved the children’s dietary knowledge…” I am not sure the data supports this claim, if so please provide support for this statement. It does support that the parents’ views on Fl and OH changed.
Overall:
I would encourage the authors to perform this study again, post-COVID, in more schools in Jordan and maybe collaborate with dental school faculty around the world to implement this in their communities. As a thought, could the came be a role play game, like Minecraft or Roblox where the person is required to eat healthy and brush their teeth and find certain things that teach them about health eating and proper oral hygiene.
Reviewer 2 Report
It is a somewhat innovative topic, with a degree of clinical relevance potentially to be explored.
In my best knowledge there was a complete compliance with the stipulated submission rules (journal-specific instructions), as far as I could verify.
The objectives are conscientiously and clearly defined, as well as the methodology in its different aspects, including statistics, as far as I also could verify. It seems to me that the methodologic steps required for this type of research have been completed, including the desirable reproducibility. I suggest perhaps changing the "t" referring to the Student´s t test to lowercase... but this is a writing detail, nothing more...
The results are properly written, in a pragmatic and well-organized manner, reflecting the methodological compliance. Perhaps the tables could be a little more enlightening in terms of data visualization, eventually the table format and the way the data is presented...
The discussion approaches, in a comparative and parallel way, the results of publications in this line of objectives, in order to obtain structured and adequately grounded conclusions. In the discussion topic, some critical sense is evident, which, in my opinion, constitutes a factor of increased interest.
The conclusions are brief, unpretentious and pragmatically based on the results.
The bibliographic references included in the sequence of the adopted methodology are in accordance with the stipulated norms.
Figures, Tables and Appendices are generally suitable in terms of content. Eventually an improvement in the tables, as mentioned, would be an important improvement, although not strictly necessary.
Reviewer 3 Report
Dear Authors,
I have carefully read and revised your work entitled “A video-game-based oral health intervention in primary schools – a Randomised Controlled Trial”.
It is well written and organized, however some points should be clarified before. When you upload the revised file, please attach a Word file so that modifications and suggestions can be written on it. Please, highlight the corrections with a color mark.
Here is a list of the abovementioned issues:
- Abstract: please delete the “trial registration” in the abstract section, as it is already displayed in the M&Ms section. Moreover, you should add that “data were submitted to statistical analysis, together with significance level (P<0.05? P<0.01? In the text, this aspect was not specified). Please, also remove (1), (2), (3) and (4) numbers.
- Introduction: the statistical null hypotheses of the study are missing. Please, add them accordingly.
- Materials and Methods: you should follow CONSORT checklist for paragraphs. An example of paragraphs should be:
- Trial design: you can add here the trial registration and the Ethical approval.
- Participants: you should describe where patients were recruited, the inclusion/exclusion criteria adopted
- Interventions and Outcomes: you should explain the study, including the number of arms, the intervention assignation, the assessed variables and so on. Moreover, you should specify the follow-up/s.
- Sample size: you should add the calculations (expected mean and SD, and expected mean difference of the primary outcome)
- Randomization and Blinding
- Statistical Methods: were data normally distributed? You can state this aspect describing the appropriate statistical test (Shapiro-Wilks?), followed by the subsequent test.
- Results: the flow chart of the study should follow CONSORT flowchart. Please, provide a new figure. The mean ages of the participants are missing. This is an important aspect. “The participants’ basic characteristics ïƒ the participants’ baseline data”. You should combine tables and add significant differences in the tables (e.g. line 203: P=0.008 I do not find this p value in the tables, please add all the data). Moreover, the descriptions of the results should be summarized and should highlight the most important findings more clearly.
- Discussion: you should discuss that children could undergo orthopedic/orthodontic treatment and this could affect the results. I suggest discussing also the use of smartphone applications in a similar way the Authors did in their study:
- https://doi.org/10.3390/app11020706
- Spelling Issues: English spelling should be improved by an English native speaker.
Round 2
Reviewer 1 Report
Thank you for your response. You have addressed all my comments and questions.
Author Response
Thank you once again for taking the time to review our manuscript
Reviewer 2 Report
I consider that, after the changes introduced in this latest version, the contents generally meet the requirements to be able, in my opinion, to be accepted for publication.
Author Response
Thank you once again for taking the time to review our manuscript.
Reviewer 3 Report
Dear Authors,
Thank you for providing the revised version of your manuscript entitled "A video game-based oral health intervention in primary schools – a Randomised Controlled Trial".
Before acceptance for publications, there are still some modifications to perform:
- sample size: when I perform the calculation, I obtain that 12 patients per group. I suggest choosing a higher SD and increasing power level to reach the right sample size.
- to simplify the reading of tables, you can write data in this way 10.5±4.2. You can avoid writing 13.4 (SD=5.4) and so on. In the table footer, you can write (mean±SD) as a legend.
Thank you for your hard work.
Round 3
Reviewer 3 Report
Dear Authors,
The manuscript is now suitable for publication.
Thank you for your hard work.
Author Response
Thank you again for taking the time to review our manuscript.